# Insulin Resistance in Women Correlates with Chromatin Histone Lysine Acetylation, Inflammatory Signaling, and Accelerated Aging

**DOI:** 10.3390/cancers16152735

**Published:** 2024-08-01

**Authors:** Christina M. Vidal, Jackelyn A. Alva-Ornelas, Nancy Zhuo Chen, Parijat Senapati, Jerneja Tomsic, Vanessa Myriam Robles, Cristal Resto, Nancy Sanchez, Angelica Sanchez, Terry Hyslop, Nour Emwas, Dana Aljaber, Nick Bachelder, Ernest Martinez, David Ann, Veronica Jones, Robert A. Winn, Lucio Miele, Augusto C. Ochoa, Eric C. Dietze, Rama Natarajan, Dustin Schones, Victoria L. Seewaldt

**Affiliations:** 1City of Hope Comprehensive Cancer Center, Duarte, CA 91010, USA; cvidal@coh.org (C.M.V.); jalvao@coh.org (J.A.A.-O.); parijatsenapati.3@gmail.com (P.S.); jtomsic@coh.org (J.T.); vrobles@coh.org (V.M.R.); cresto@coh.org (C.R.); nancsanchez@coh.org (N.S.); angsanchez@coh.org (A.S.); nemwas@coh.org (N.E.); daljaber@coh.org (D.A.); dann@coh.org (D.A.); vjones@coh.org (V.J.); edietze@coh.org (E.C.D.); 2Arthur Riggs Diabetes and Metabolism Research Institute, City of Hope Duarte, Duarte, CA 91010, USA; zhuchen@coh.org (N.Z.C.); nbachelder@coh.org (N.B.); rnatarajan@coh.org (R.N.); 3Sidney Kimmel Cancer Center, Philadelphia, PA 19107, USA; terry.hyslop@jefferson.edu; 4Department of Biochemistry, University of California at Riverside, Riverside, CA 92521, USA; ernestm@ucr.edu; 5Massey Comprehensive Cancer Center, Virginia Commonwealth University, Richmond, VA 23298, USA; robert.winn@vcuhealth.org; 6School of Medicine, Louisiana State University, New Orleans, LA 70112, USA; lmiele@lsuhsc.edu (L.M.); aochoa@lsuhsc.edu (A.C.O.)

**Keywords:** epigenetics, chromatin acetylation, histone H3K9ac, insulin resistance, tissue inflammation, interlukin-6 (IL6), TNF-alpha, senescence, SASP

## Abstract

**Simple Summary:**

There is increasing evidence that Type-2 diabetes and pre-diabetes may increase cancer risk; however, the biology that links these diseases is poorly understood. Epigenetics is the biological study of how DNA can be modified without mutations; epigenetic changes link medical, social, and environmental changes with cancer. Here, we investigated whether insulin resistance (pre-diabetes) may result in epigenetic changes that increase cancer risk. We discovered that women with insulin resistance have epigenetic changes that increase inflammation and perhaps accelerated aging. Our study is important because the inflammatory changes we see are associated with an increased risk for heart disease, kidney disease, and cancer.

**Abstract:**

Background: Epigenetic changes link medical, social, and environmental factors with cardiovascular and kidney disease and, more recently, with cancer. The mechanistic link between metabolic health and epigenetic changes is only starting to be investigated. In our in vitro and in vivo studies, we performed a broad analysis of the link between hyperinsulinemia and chromatin acetylation; our top “hit” was chromatin opening at H3K9ac. Methods: Building on our published preclinical studies, here, we performed a detailed analysis of the link between insulin resistance, chromatin acetylation, and inflammation using an initial test set of 28 women and validation sets of 245, 22, and 53 women. Results: ChIP-seq identified chromatin acetylation and opening at the genes coding for TNFα and IL6 in insulin-resistant women. Pathway analysis identified inflammatory response genes, NFκB/TNFα-signaling, reactome cytokine signaling, innate immunity, and senescence. Consistent with this finding, flow cytometry identified increased senescent circulating peripheral T-cells. DNA methylation analysis identified evidence of accelerated aging in insulin-resistant vs. metabolically healthy women. Conclusions: This study shows that insulin-resistant women have increased chromatin acetylation/opening, inflammation, and, perhaps, accelerated aging. Given the role that inflammation plays in cancer initiation and progression, these studies provide a potential mechanistic link between insulin resistance and cancer.

## 1. Introduction

The mechanistic links between metabolic health and cancer risk are still relatively unexplored. The key to understanding these links is the study of epigenetics. Epigenetics refers to potentially heritable changes in gene expression that occur without alterations in the underlying DNA sequence [1]. Epigenetic dysregulation has been recently shown to play an important role in linking poor metabolic health with cancer-promoting pathways, such as inflammation and accelerated aging [1].

During Type-1 diabetes (T1DM), the insulin-producing pancreatic beta-cells undergo catastrophic cell death, and the individual with T1DM is committed to receiving life-long insulin replacement [2]. In contrast, Type-2 diabetes (T2DM) occurs slowly over time. The first step in the development of T2DM is pre-diabetes, or insulin resistance [2]. During insulin resistance, the body, primarily muscle cells, becomes progressively resistant to insulin [3]. Consequently, in an attempt to regulate normal plasma glucose levels, the beta-cells of the pancreas secrete higher and higher amounts of insulin (insulin-rich state) [4]. After insulin-resistant individuals eat, plasma insulin levels can spike to 5–20 times higher than normal [5]; this high insulin production demand progressively “burns” out the pancreatic beta-cells, and the individual subsequently develops T2DM. When T2DM occurs, insulin demand outstrips production, and plasma glucose rises (glucose-rich state).

Many well-powered studies have investigated the link between obesity and premenopausal breast cancer [3]. These studies, while extremely well designed, have led to a confusing array of results, and for the most part have been negative [3]. One potential confounder is that it is now known that individuals can be (1) obese and metabolically healthy or, conversely, (2) lean and metabolically unhealthy [6]. Consequently, there has been recent interest in studying the link between metabolic dysregulation and cancer [6].

In our previously published studies [7,8], we tested in vitro and in vivo for the impact of high insulin treatment (hyperinsulinemia) on histone acetylation in breast cancer cell lines and normal mammary epithelial cells. Chromatin immunoprecipitation (ChIP) studies identified that hyperinsulinemia led to (1) a global increase in histone acetylation—with our top “hit” at H3K9—and (2) activation of the PI3K/AKT/mTOR pathway [7,8]. Genome-wide analyses revealed that hyperinsulinemia increased histone acetylation [7,8]. In addition, hyperinsulinemia resulted in increased production of reactive oxygen species and DNA damage foci in cells [7,8].

Building on our in vitro and in vivo mechanistic studies, we investigated whether the results of our preclinical, mechanistic studies demonstrating chromatin acetylation at H3K9 were similarly observed in women with pre-diabetes or insulin resistance. When an individual with insulin resistance eats, as discussed above, they produce abnormally high insulin levels. High insulin is known to overdrive the mitochondrial electron transport chain and result in the overproduction of bioactive metabolites, including acetyl-CoA [3]. Recent evidence shows that the overproduction of acetyl-CoA results in the hyper-acetylation of histone proteins and increased chromatin accessibility [6]. Chromatin accessibility is a characteristic feature of regulatory regions in eukaryotic genomes, and alterations to chromatin accessibility impact gene regulation [7,8].

Here, we report that, consistent with our preclinical studies, insulin resistance is associated with the acetylation of H3K9 in peripheral blood mononuclear cells (PBMCs) of women with insulin resistance. This, in turn, promotes cytokine production/inflammation (interleukin-6, IL6; tumor necrosis factor-alpha, TNFα; and cytokine signaling) and cellular senescence (NFκB-signaling and innate immunity). Using methylation assays, we also observed that insulin resistance is potentially associated with accelerated aging in postmenopausal women.

Cellular senescence is defined as terminal G1/S arrest that occurs in response to telomere shortening, DNA damage, and inappropriate expression of oncogenes [9,10]. Senescence was first described as a pathway to prevent cancer [9,10]. However, there is growing evidence that senescent cells may also contribute to cancer due in part to the generation of inflammatory senescence-associated secretory phenotype (SASP) cells [9,10]. SASP cells produce IL6 and TNFα [9,10], which, in turn, activate IL6/TNFα/NFκB-signaling networks. We observed that these networks are in circulating PBMCs in women with insulin resistance.

Until recently, PBMCs were not thought to be able to actively impact senescence. A recent study [11], however, showed that senescent PBMCs could cause senescence in healthy solid organs. The transplantation of senescent bone marrow from genetically engineered senescent mice into recipient syngeneic healthy wild-type mice resulted in senescence and premature aging of the previously healthy recipient’s solid organs including the heart, kidney, and liver. These data show that a senescent immune system has a causal role in driving systemic aging [11]. Recently, senescence has also been shown to impact diabetes in animal models; treatment with drugs that remove senescent cells leads to improvements in blood glucose levels and a reduction in diabetic complications [12]. In our published work, we showed that women with germline *BRCA* mutations have accelerated aging [13]. Given the important role that inflammation has on cancer initiation and progression, our findings provide a potential link between insulin resistance and cancer.

## 2. Materials and Methods

### 2.1. Human Subjects

Women consented to City of Hope Institutional Review Board-approved COH Protocol#/Ref#: 18306/158149. This protocol was approved by the Human Subjects Committee and the Institutional Review Board at City of Hope. Women consented either in person or by telemedicine and gave permission for our team to access the woman’s City of Hope medical record. Women were required to not have a prior diagnosis of (1) invasive cancer or (2) T2DM and (3) were required to not be a smoker. Women provided demographic information including age, self-identified race, and medical history. Weight and height were provided by the women’s medical records. Women completed a survey that included information on diet, transportation, housing, and medical history.

### 2.2. Human PBMC Isolation

Blood samples were obtained from women, following institutional guidelines at City of Hope (IRB no. 18306), in purple-top EDTA vacutainer tubes (Avantor, Radnor Township, PA, USA, Catalog No. 366643) after obtaining written informed consent. Samples were first diluted with equal volumes of wash buffer (phosphate-buffered saline (PBS) containing 2% fetal bovine serum (FBS)) and subsequently layered into SepMate tubes (StemCell Technologies, Vancouver, BC, Canada, Catalog No. 85450) preloaded with 15 mL Ficoll-Paque (GE Healthcare, Anaheim, CA, USA, Catalog No. 17-5442-02). Samples were then centrifuged at 1200× *g* for 10 min at room temperature (RT, 15–25 °C). The upper plasma layer was then removed without disturbing the plasma:Ficoll interface that contains the buffy coat with the PBMCs. Three volumes of wash buffer were then added into the PBMC buffy coat layer and centrifuged at 300× *g* for 8 min at RT. Cells were then washed with 10 mL wash buffer and then centrifuged at 100× *g* for 10 min at RT. The cells were then resuspended in 5 mL of wash buffer and were counted using the Cellometer Auto 2000 Cell Viability Counter (Nexcelom Biosciences, San Diego, CA, USA). For ChIP experiments, PBMCs (5.0 × 10^6^) were crosslinked using 1% formaldehyde for 10 min at RT, followed by the addition of 0.125 M glycine for 5 min to stop the reaction. Crosslinked cells were then washed twice with ice-cold PBS. PBMC pellets were stored at −80 °C. For flow cytometry, PBMCs were resuspended in RPMI containing 45% FBS and 10% dimethyl sulfoxide and stored at −196 °C.

### 2.3. ChIP-Rx

ChIP-Rx was performed as previously described [7,8] with minor modifications. After PBMC isolation, PBMC pellets were resuspended in SDS lysis buffer (1% SDS, 10 mM EDTA, 50 mM Tris-HCl, pH 8) supplemented with protease inhibitors and subjected to sonication using a BioruptorR pico (Diagenode, Denville, NJ, USA) for six cycles (30 s on/30 s off) to produce DNA fragments of 200–500 bp in length.

ChIP-seq libraries were made using the Illumina Tru-Seq library preparation kit (Illumina, San Diego, CA, USA) and multiplexing barcodes compatible with Illumina HiSeq 2500 technology. About 50 million single-end reads of length 51 bp were generated from each ChIP-seq library.

### 2.4. ChIP-Seq Analyses

Sequencing reads from each library were trimmed to remove low-quality reads and adapters using Trim Galore [14], version 0.4.5. Trimmed reads were aligned to the human hg19 reference genome using Bowtie [15]. Bowtie (v1.1.2) alignment was performed using the following parameters: -e 70 -k 1 -n 2-best-chunkmbs 200. Picard tools were used to remove PCR duplicates. Bigwig files were generated from bam files using bedtools genomecov [16] and UCSC tools. Read counts were normalized by the number of reads mapping to the exogenous reference genome (dm3) [17]. Bigwig files were visualized on the UCSC Genome Browser [18].

To identify genes with significant increases in promoter H3K9ac signal, aligned read counts were obtained for promoters of all RefSeq genes (TSS ± 2 kb) for each sample using bedtools coverage. Subsequently, DESeq2 [19] was used to compare insulin-sensitive and insulin-resistant samples to identify promoters with significant changes in the H3K9ac signal. A total of 476 gene promoters with a significant increase in the H3K9ac signal were identified (log2foldchange > 1 and FDR < 10^−8^). Aggregate H3K9ac ChIP-seq profiles were made using deepTools2 [20].

### 2.5. Gene Set Enrichment

Gene sets enriched among these genes were identified using the enricher function in ClusterProfiler [21]. Gene sets within Hallmark (H), curated (C2), and ontology (C5) collections of the Molecular Signatures Database (MSigDB) were retrieved using the Msigdbr package. A hypergeometric test was used to identify enriched gene sets (adj. *p*-values < 0.05). A gene set enrichment plot was made using ggplot2 (version 3.5.1). The Cnetplot function within ClusterProfiler was used to generate a visualization of genes involved in selected pathways as indicated.

### 2.6. Motif Enrichment

Motif enrichment analysis was performed using findMotifsGenome.pl in Homer 4.10 [21]. Promoter sequences (TSS ± 2 kb) of genes with the highest increases in the H3K9ac signal were used to perform the motif analyses. For each set of promoter regions, background sequences with matched GC content were selected, and *p*-values for motif enrichment were calculated using cumulative binomial distributions. Figures were made using R (version 3.5.1) and Adobe Illustrator.

### 2.7. Cytokine Arrays

Cytokine arrays were performed using Luminex High Performance Panels (R&D Systems, Minneapolis, MN, USA) as per the manufacturer’s instructions by the Analytical Pharmacology Shared Resource at the City of Hope Comprehensive Cancer Center. Cytokines were tested in triplicate. Cytokines tested were TNFα, IL6, IL15, IL16, IL23, Leptin, and CXCL1.

### 2.8. Senescence Assay in Circulating Lymphocytes Using Flow Cytometry

PBMCs (1.0 × 10^6^) were prepared for flow cytometry by staining for the immune cell subsets, CD45+CD3+CD4+ T-cells, and CD45+CD3+CD8+ T-cells using anti-human CD45 V500 (BD Biosciences, Frankliin Lakes, NJ, USA, Catalog No. 560777), anti-human CD3 Super Bright 600 (ThermoFisher Scientific, Waltham, MA, USA, Catalog No. 63-0037-42), anti-human CD4 PE-Cy7 (BD Biosciences, Catalog No. 348789), and anti-human CD8 cFlourYG610 (Cytek Biosciences, Fremont, CA, USA, Catalog No. R7-20245). Dead cells were excluded using Zombie NIR (Biolegend, San Diego, CA, USA, Catalog No. 423106). The CellEvent Senescence Green Flow Cytometry Assay Kit (ThermoFisher, Catalog No. C10841) was used to detect senescence-associated β-galactosidase (SA-β-gal) activity. Following cell surface receptor staining, cells were fixed with 4% paraformaldehyde for 15 min, washed using PBS containing 0.5% bovine serum albumin and 5 mM EDTA, and incubated with the CellEvent senescence green probe (1 to 1000 dilution) in CellEvent senescence buffer for 2 h at 37 °C. Samples were analyzed on a Cytek Aurora (100,000 events), and data analysis was performed using Cytek SpectroFlo software (version 3.2.1).

### 2.9. DNA Methylation DNAme Profiling and Data Preprocessing

Genomic DNA was isolated from whole blood using the QuickGene DNA Whole Blood Kit (Autogen, Holliston, MA, USA). DNA non-methylated cytosine residues were converted into uracil using the EZ DNA Methylation Kit (Zymo Research, Irvine, CA, USA) with an alternative incubation for the Illumina HD Infinium Methylation Assay (Illumina, San Diego, CA, USA). The Infinium MethylationEPIC v2.0 Kit (Illumina) was used according to the manufacturer’s protocol. Briefly, bisulfite-converted DNA was denatured using 0.1 N NaOH and amplified for 20–24 h at 37 °C in a hybridization chamber. The amplified DNA was enzymatically fragmented for 1 h at 37 °C and then precipitated for 30 min at 4 °C. DNA hybridization onto the Illumina beadchip was performed by resuspending precipitated DNA using RA1 solution and loading it onto the Infinium Methyl EPIC 8 sample beadchip (Illumina). Hybridization was performed for at least 16 h and completed within 24 h. Image acquisition was carried out using the Illumina iScan System.

For each sample, analyses on quality control were performed, including evaluating EPIC array internal controls, examining genome-wide DNAme level distributions, and calculating the percentage of unreliable CpGs with detection *p* > 0.05. After ensuring all of the samples were of good quality, normal-exponential out-of-band normalization [22] was applied to each sample followed by between-array stratified quantile normalization [23] to obtain a normalized dataset. CpGs with detection *p* > 0.05 in at least one sample and CpGs containing SNPs in the probe regions were removed from the dataset, resulting in a DNAme dataset in the format of beta values (defined as the ratio of methylated signals vs. combined intensity of both methylated and unmethylated signals) at 905,228 out of 936,990 probes covered by the EPIC v2 array. The dataset was further summarized by calculating mean values on CpGs with replicated probes for each sample to obtain CpG-level DNAme of 899,207 CpGs across 54 samples. Sample-level cluster analysis on the dataset did not reveal any outlier(s) or sample clusters based on slides. Hence, the dataset of all 54 samples is considered a reliably measured dataset for subsequent analyses. All of these analyses were performed using functions with default parameters provided in the R Bioconductor package Minfi v1.48.0 [24].

### 2.10. Assignment of Disparities Index

The disparity index was measured using the Area Deprivation Index (ADI) determined at the neighborhood level [25,26]. The ADI is a validated composite ranked index containing 17 census block groups of (i.e., “neighborhood-level”) social determinants of health factors encompassing housing, income, employment, transportation, and education as captured in the American Community Survey. The ADI state rankings range from 1 to 10, with the least disadvantaged neighborhood conditions designated by lower scores and the most disadvantaged by higher scores. The patients’ 9-digit zip codes were geolinked to their ADI neighborhood ranking matched to the period of study.

### 2.11. Statistics

Univariate analysis was performed using simple statistical methods. Normalcy was tested using the Shapiro–Wilk and Kolmogorov–Smirnov analytic methods. Survey data and cytokine data were merged based on patient ID. Cytokine expression was log-transformed to provide a symmetric distribution for analysis. Spearman correlations of cytokines with HbA1c were computed using the cor procedure in R and plotted using the corrplot function. *p*-values assess the strength of the correlation, and we highlight only those with p < 0.001 to be conservative due to multiple tests. An additional analysis using linear mixed models assessing the association of each cytokine with HbA1c was also completed, with results analogous to Spearman correlations. GraphPad Prism version 10 software was used to perform statistical analysis of flow cytometry data.

### 2.12. DNAme Age Analysis

The normalized DNAme was used to estimate DNAme age using R package methylclock_v1.8.0 [27]. Among the DNAme clocks covered in methylclock, only the ones developed using adult blood samples with less than 5% prediction-required CpGs missing in our DNAme dataset were considered candidate clocks in the current study. To choose an optimal clock, we first estimated the DNAme age of each subject in the healthy control group for each candidate clock and then selected the clock with the highest Pearson’s correlation between the predicted DNAme age and chronological age across the 26 healthy controls. Using the selected optimal clock, DNAme age was estimated for each subject. The correlation of DNAme age with chronological age in the healthy control group (control) or insulin-resistant group (IR) was analyzed by Pearson’s correlation analysis. Age accelerations were subsequently calculated for each subject as the residual obtained after regressing chronological age on DNAmeAge adjusted for cell compositions across all the subjects. Two-sided student *t*-tests were used to compare the age accelerations between IR vs. control groups.

## 3. Results

### 3.1. Insulin Induces H3K9 Acetylation on Gene Promoter Regions Involved in Inflammatory Signaling

In our published in vitro and in vivo studies, we identified that the treatment of human cell lines and mice with insulin promoted chromatin opening; our top modified chromatin “mark” was H3K9ac [7,8]. To validate our preclinical models, we performed a detailed analysis to test the impact of insulin resistance on chromatin opening on 28 women. There were 13 insulin-resistant (HbA1c 5.7–6.3) women versus 15 metabolically healthy (HbA1c < 5.7) women (Table 1). The mean age was slightly higher in insulin-resistant women, at 61 years, vs. metabolically healthy women, at 57 years, but the difference was not statistically significant (*p* = 0.24). The mean body mass index (BMI) was slightly higher in insulin-resistant women at 27 kg/m^2^ vs. 25 kg/m^2^ in metabolically healthy women, but the difference was not statistically significant (*p* = 0.32).

We performed H3K9ac ChIP-seq on PBMCs. The aggregate H3K9ac signal was higher across 476 gene promoter regions in insulin-resistant women compared to metabolically healthy women with a normal HbA1c. Our top two differentially H3K9ac genes coded for the inflammatory cytokines tumor necrosis factor-alpha (TNFα) and interleukin-6 (IL6) (Figure 1A,B).

Pathway analysis compared genes with increased H3K9ac promoter signal in the 13 insulin-resistant women versus 15 metabolically healthy controls. Genes with the highest increase in H3K9ac signal in insulin-resistant versus healthy control women were identified by DESeq2; log2FC > 1 and FDR < 10^−8^ (Figure 1C). A total of 476 genes were identified. The top 20 enriched genes included genes involved in TNFα signaling, immune signaling, and cytokine production (Figure 2A,B). Pathway analysis identified inflammatory response genes, NFκB/TNFα-signaling, inflammation, reactome cytokine signaling, and innate immunity (Figure 2A,B).

### 3.2. Cytokine Analysis

Our ChIP-seq and pathway analysis provided evidence for inflammation and cytokine signaling in insulin-resistant women versus healthy controls. Cytokine arrays in triplicate tested for the presence of inflammatory cytokines. Our analysis included TNFα, IL6, and Leptin. Insulin-resistant women (*n* = 13), versus metabolically healthy controls (*n* = 15), had significantly increased mean serum levels of TNFα of 6.34 units (interquartile range 4.97–8.68) vs. 3.20 units (interquartile range: 1.81–4.21), *p* = 0.0012; IL6 of 2.62 units (range 1.25–3.95) vs. 0.24 units (interquartile range: 0.25–1.68), *p* = 0.0007; and leptin of 52,300 units (interquartile range: 5849–88,100) vs. 1706 units (interquartile range: 2524–24,740), *p* = 0.010 (Figure 3). These data provide evidence that acetylation at H3K9ac and chromatin opening of the genes coding for TNFα and IL6 is correlated with increased cytokine levels.

### 3.3. Transcription Factor Motif Analysis

We next tested for transcription factor motifs enriched at promoters with increased H3K9ac signal in our insulin-resistant women (*n* = 13) vs. metabolically healthy women (*n* = 15). Transcription factor motifs that were enriched included interferon-regulatory factor-1 (IRF1), NFκB, and runx-related transcription factor-1 RUNX1 (Figure 4).

### 3.4. Expanded Analysis of HbA1c and Cytokines

Our initial analysis of 28 insulin-resistant women and healthy controls provided evidence of increased chromatin acetylation at H3K9 for genes coding for the inflammatory cytokines TNFα and IL6. We also performed cytokine analysis on 245 women (Table 2 below). Cytokine arrays tested for TNFα, IL6, IL15, IL16, IL23, leptin, and CXCL1; associations were assessed via Spearman’s correlation (Figure 5 below). TNFα and leptin were significantly positively associated with HbA1c, while CXCL1 was significantly negatively associated with HbA1c. Also, note the significant correlation of TNFα to IL15, IL6, leptin, and IL23 as well as the strong positive association of leptin and IL23.

### 3.5. Insulin Resistance and Senescence

Cellular senescence has been shown to play a key role in T2DM and its complications [12]. In our cohort of insulin-resistant women, the ChIP-seq and cytokine array data found activation of the IL6/TNFα/NFκB-signaling networks, which are known to play a role in cellular senescence. Therefore, we investigated whether there was a senescence-associated phenotype using flow cytometry in 22 additional women ranging from 34 to 62 years of age. We analyzed the T-cells of insulin-resistant (*n* = 9) and metabolically healthy women (*n* = 13) for senescence-associated β-galactosidase (SA-β-gal) activity, a marker of senescent cells (Table 3 below). The mean age was slightly higher in insulin-resistant women, at 53 years (range 37 to 56 years), vs. metabolically healthy women, at 49 years (range 45 to 61 years); this difference was not significant *p* = 0.053. There was a significant increase in the percentage of senescent CD4+ and CD8+ T-cells in insulin-resistant women vs. metabolically healthy controls; *p* = 0.0046 and *p* = 0.0245, respectively (Figure 6 below). One-way ANOVA was used to determine significance.

### 3.6. Methylation Measures of Aging

Our pathway analysis and flow cytometry analysis of T-cells (Figure 6 above) provide evidence that insulin resistance may, perhaps, be associated with accelerated aging. To evaluate this possibility, we performed methylation measures of aging, or “methylation clocks”, in 53 women. We extracted DNAs from whole blood samples of healthy controls (control group *n* = 25) and insulin-resistant women (IR group *n* = 28). Control and insulin-resistant women were balanced for BMI (median = 30 vs. 28), and age was slightly higher in the control group (median = 54 vs. 51). We profiled DNAme using the human Infinium MethylationEPIC v2.0 array. Using the resulting normalized DNAme dataset containing DNAme levels in beta values at 899,207 CpGs reliably measured across all 53 women (see Methods), we predicted each subject’s DNAme age using published DNAme clocks to evaluate whether insulin resistance would impact aging. To choose the best DNAme clock, we identified four candidate DNAme clocks, which were generated using similar samples to our study (adult blood samples) and have less than 5% missing CpGs required for age prediction for the clock in our dataset (Table 4 below). We then applied each clock to the normalized DNAme dataset and calculated the DNAme age for each healthy control. Zhang’s clock was finally chosen as the optimal DNAme clock to predict DNAme age in our study considering that the DNAme age estimated by this clock showed the highest correlation with chronological age (*r* = 0.95, *p* = 2.31 × 10^−13^.

The DNAme age of all subjects in the insulin-resistant group was estimated using Zhang’s clock. A high correlation between DNAme age and chronological age ≥55 and <55 across samples in the insulin-resistant group (*r* = 0.93, *p* = 1.87 × 10^−12^) was also observed. But the level of correlation in the insulin-resistant group is mildly smaller than that in the control group, suggesting that insulin resistance might potentially impact aging.

To further evaluate this effect, we calculated age acceleration for each subject, which captured the difference between DNAme and chronological age after adjusting for cell composition. We also categorized the cohort into two sub-cohorts using chronological ages. One sub-cohort contained 29 women with a chronological age ≤55 (younger sub-cohort), which included 17 subjects in the control group and 12 in the insulin-resistant group. In this younger sub-cohort group, the mean age was balanced between age (51 vs. 50 years, respectively) and BMI (29 vs. 29, respectively).

The second sub-cohort contains 24 women with an age >55 (older sub-cohort), including 8 in the control group (metabolically healthy) and 16 in the insulin-resistant group. In this older sub-cohort group, mean age was balanced between age (61 vs. 61 years, respectively) and slightly higher in the metabolically unhealthy group for BMI (28 vs. 30, respectively).

We compared the age acceleration between control (metabolically healthy) and insulin-resistant groups in both sub-cohorts. We detected a trend of increased age acceleration in the insulin-resistant vs. control group in the older sub-cohort (age >55), with a mean value of 0.05 in the insulin-resistant vs. −1.48 in the control group at a marginal *p* = 0.10 (Figure 7A below), while no significant difference was detected in the younger sub-cohort, *p* = 0.73 (Figure 7B below). This result not only indicated that insulin resistance might accelerate aging in older populations but also provided evidence that maintaining a healthy blood glucose level could prevent aging with a negative age acceleration.

## 4. Discussion

Epigenetics provides a mechanism by which poor metabolic health can lead to inflammation, accelerated aging, and, ultimately, disease [28]. In our published mechanistic studies [7,8], we identified in vitro and in vivo that hyperinsulinemia led to a global increase in histone acetylation. Our top “hit” was at H3K9 [7,8]. Consistent with these mechanistic studies, here, we observed that insulin-resistant women had increased chromatin acetylation at H3K9. Our top two genes with increased histone acetylation coded for the inflammatory factors TNFα and IL6. Consistent with this finding, insulin-resistant women had increased TNFα/IL6 cytokine levels. Detailed acetylation studies were performed in a relatively small sample set (*n* = 28). However, cytokine results were validated in an expanded set of 245 women. Taken together, these data provide an epigenetic link between insulin resistance and inflammation.

In insulin-resistant women (vs. metabolically healthy controls), we identified transcription factor motifs with increased H3K9ac promoter signal. IRF1 and NFκB are of particular interest because they are involved in interferon-beta signaling and recruitment of the platform for the PCAF chromatin modification complex and *p*300/CBP acetyltransferase [29]. NFκB is also of interest because of its role in cellular senescence [30].

Our studies provide evidence that individuals with insulin resistance (versus metabolically healthy individuals) have increased (1) production of IL6 and TNFα and (2) percentages of senescent CD4+ and CD8+ T-cells, consistent with an inflammatory senescence-associated secretory phenotype (SASP). SASP is known to play a central role in many diseases, such as T2DM, cardiovascular disease, and diabetic kidney disease. There is increasing evidence that SASP may also contribute to cancer initiation and progression [9,10].

SASP cells are thought to contribute to tumorigenesis via the production of inflammatory cytokines, IL6 and TNFα, and the creation of an inflammatory microenvironment [10]. In the microenvironment, inflammatory SASP cells have been shown to promote (1) cancer angiogenesis, invasion, and metastasis [9,10]; (2) “stemness”; and (3) metabolic reprogramming [31,32]. We also see evidence of epigenetic aging in insulin-resistant women, but our findings are statistically significant only in postmenopausal women. Taken together, our studies provide a potential mechanistic link between insulin resistance, chromatin acetylation, and accelerated aging. Given the impact of aging and accelerated aging, particularly in women with germline *BRCA* mutations [13] on cancer, these studies provide a link between insulin resistance, accelerated aging, and cancer.

Until recently, circulating WBCs were not thought to be able to drive senescence in distal organs such as the pancreas or heart [11]. A recent study [11], however, showed that transplanted senescent WBCs could cause senescence in healthy solid organs. These data show that a senescent immune system can have a causal role in driving systemic aging [11]. Our studies provide a potential link between insulin resistance, inflammatory circulating WBCs, and, perhaps, accelerated aging. We observed potential evidence of accelerated aging by flow cytometry analysis in T-cell subsets and methylation markers and potential evidence of accelerated aging in postmenopausal women. Methylation analysis of the association between premenopausal women and accelerated aging was not statistically significant; however, our dataset was relatively small, and we are performing additional analysis of premenopausal women.

This study is limited to bulk peripheral blood cells; it is not clear which cells are responsible for the epigenetic and inflammatory changes. Second, we did not correlate changes in peripheral blood cells with cells in organs such as the heart or kidney. However, these correlations have been seen in animal models.

Alterations to chromatin acetylation provide a molecular mechanism by which insulin resistance and declining metabolic health can contribute to epigenetic changes and, ultimately, disease. In our current and future studies, we are testing if metabolically unhealthy women who become metabolically healthy can reverse the epigenetic and aging changes we observe to be associated with insulin resistance or whether these changes are permanent. The answer to this question is important for public health and disease prevention.

## 5. Conclusions

The link between insulin resistance, epigenetic damage, and disease has only just started to be investigated. Our studies are limited in size and need to be validated in a second cohort and more broadly in national studies. We feel, however, that with these validation steps, our findings have the potential to provide mechanistic risk biomarkers to assess the impact of early detection and treatment of pre-diabetes/insulin resistance on cancer prevention. Previous studies have identified links between adverse social determinants of health and elevated inflammatory cytokines such as IL6 and TNFα (for a review, see reference [28]). While our studies here are limited in their scope, they lead us to hypothesize that there may be common disease risk pathways. We speculate that adverse social determinants of health, together with metabolic dysregulation, (1) may have common biologic pathways, such as elevation of inflammatory cytokines and accelerated aging, that (2) increase the risk for many diseases, including cardiovascular disease, diabetes, and kidney disease. We aim, in future studies, to test for these common pathways and, hopefully, bridge the gap between social determinants of health.

## Figures and Tables

**Figure 1 cancers-16-02735-f001:**
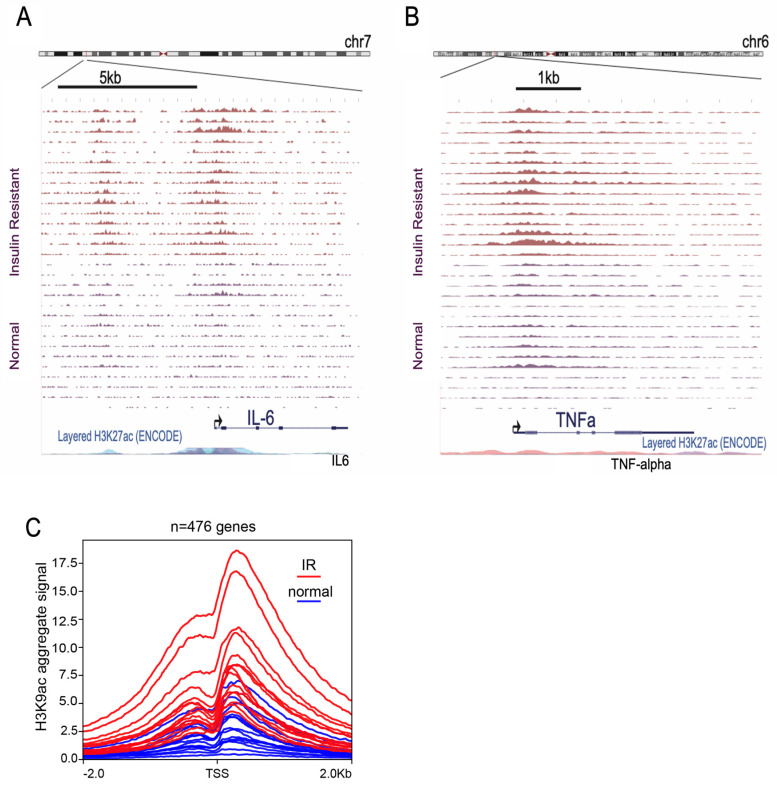
Peripheral blood mononuclear cells from insulin-resistant women have global increase in H3K9ac. (**A**,**B**) Genome browser screenshots of H3K9ac ChIP-seq data for insulin-resistant (IR) and healthy normal control women of the IL6 (**A**) and TNFα (**B**) gene locus. (**C**) An aggregate plot of H3K9ac for all 30 samples at the 476 gene promoters with increased H3K9ac.

**Figure 2 cancers-16-02735-f002:**
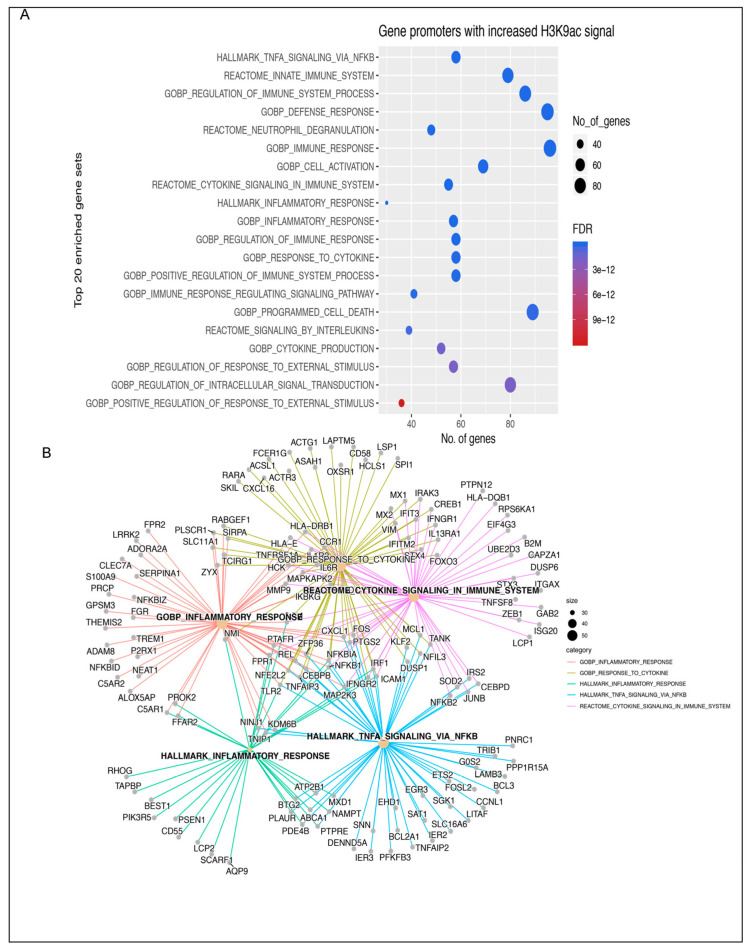
Genes with increased H3K9ac are involved in inflammatory pathways. (**A**) Gene set enrichment analysis revealed pathways of NFκB/TNFα-signaling, inflammation, reactome cytokine signaling, and innate immunity. (**B**) Many other genes involved in these pathways are potentially impacted as well.

**Figure 3 cancers-16-02735-f003:**
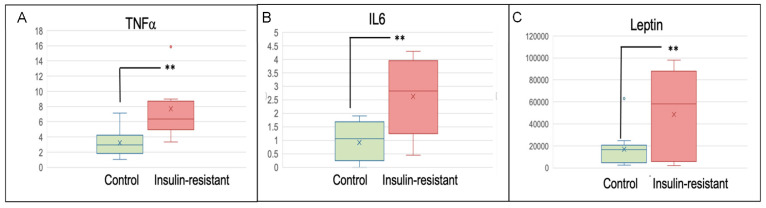
Cytokine array analysis of serum from insulin-resistant women (*n* = 13) vs. healthy controls (*n* = 15) tested in Figure 1 and Figure 2. (**A**) TNFα, (**B**) IL6, and (**C**) leptin. Samples run in triplicate relative to standard curve. Units are relative to standard curve. Outlier data shown. x is average value. ** *p* ≤ 0.01.

**Figure 4 cancers-16-02735-f004:**
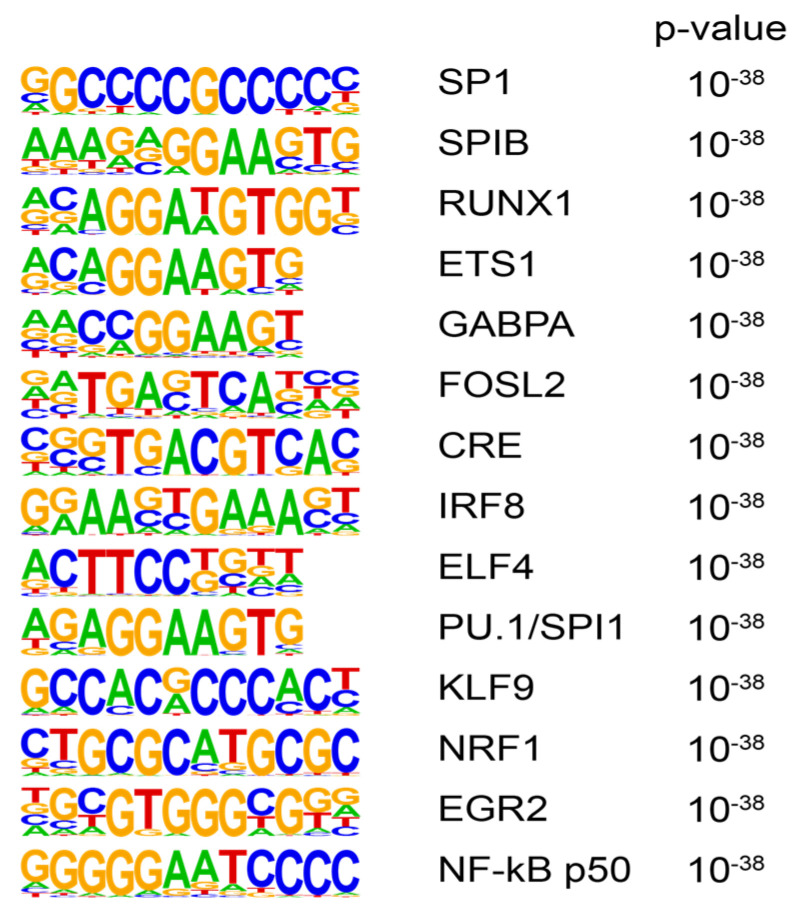
Transcription factors RUNX1, IRF1, and NFκBp50 are involved in H3K9ac chromatin remodeling in insulin-resistant women. Transcription factor-binding motifs enriched in promoter regions (TSS ± 2 kb) of upregulated genes.

**Figure 5 cancers-16-02735-f005:**
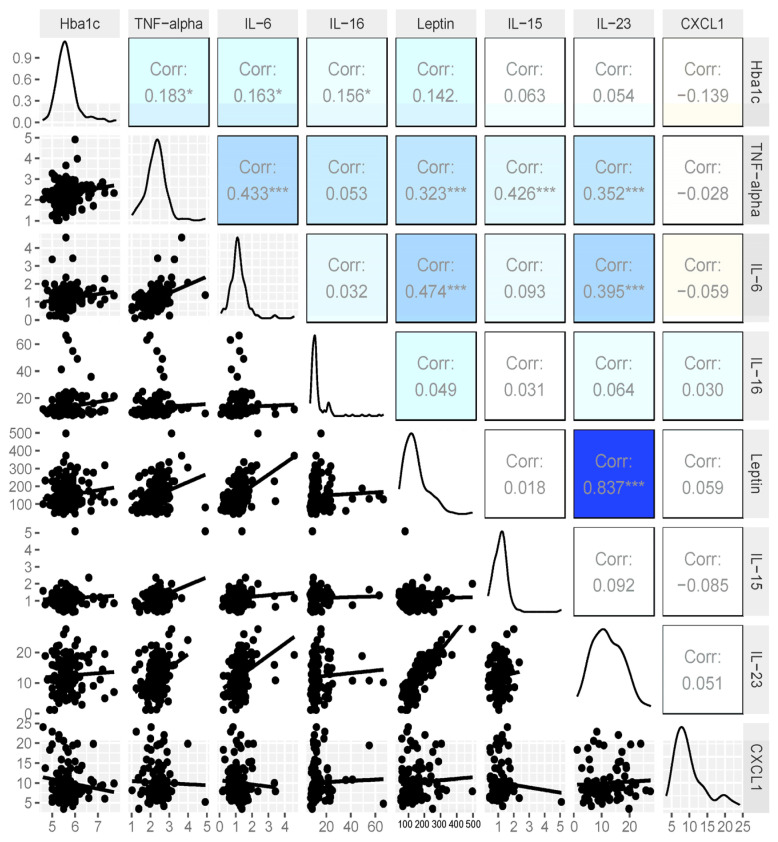
Scatter plots and Spearman correlation values between HbA1c and cytokines. TNF-alpha and leptin significantly positively associated with HbA1c (all *p* < 0.001): *** *p* <0.001, * *p* < 0.05.

**Figure 6 cancers-16-02735-f006:**
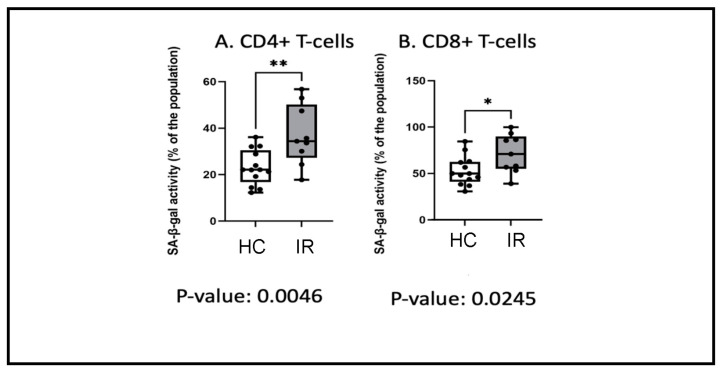
Increased senescent T-cells in insulin-resistant (**IR**) women versus metabolically healthy controls (**HC**). Flow cytometry was used to test peripheral white blood cells (WBCs) from 9 insulin-resistant women (HbA1c = 5.7–6.3) vs. 13 metabolically healthy women (HbA1 < 5.7) (Table 3) for beta-galactosidase (SA-beta-gal) activity, a marker of senescence. There was a significant increase in the percentage of senescent CD4+- and CD8+-positive T-cells in insulin-resistant women vs. healthy women; ** *p* = 0.0046 and * *p* = 0.0245, respectively.

**Figure 7 cancers-16-02735-f007:**
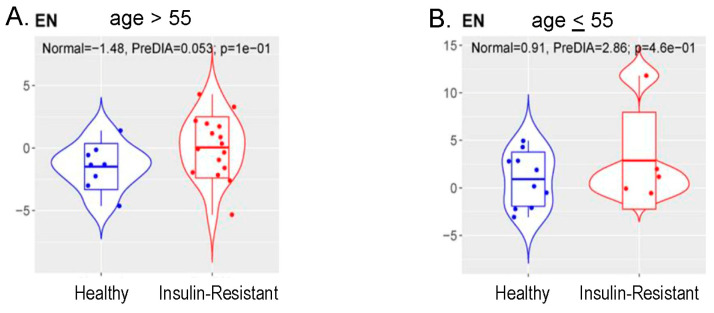
Correlation between insulin resistance and methylation measures of aging. DNAme age of all groups was estimated using Zhang’s clock. In women with age >55 (**A**), insulin resistance showed higher age acceleration estimated by DNA methylation compared to metabolically healthy (Healthy) group (mean 0.053 vs. −1.48) at marginal *p* = 0.10 (two-sided *t* test). The association in women ≤ 55 (**B**) was not significant, but the number of insulin-resistant women aged ≤ 55 was small.

**Table 1 cancers-16-02735-t001:** Impact of insulin resistance on chromatin opening; analysis of 28 women.

HbA1c	Age	Race	Ethnicity	BMI (kg/m^2^)
**Metabolically Healthy**				
5.3	54	Asian		24
5.4	59	Asian		20
5.2	64	Black/Indigenous		24
5.4	60	not specified	Latina	31
5.5	60	White		29
5.5	59	White		26
5.6	61	White	Latina	26
5.1	65	White	Latina	24
5.6	54	Asian		25
5.6	66	White		27
5.6	31	White		23
5.2	64	White	Latina	31
5.1	45	White		20
5.3	54	Asian		24
5.4	59	Asian		20
**Insulin Resistant**				
5.8	59	White	Latina	33
5.9	66	White		29
5.8	56	White		24
5.7	58	Black		24
5.7	73	White		34
5.8	48	Asian		24
6.1	61	Black		31
6.0	58	Black		19
6.0	66	White		26
5.8	50	Asian		21
5.7	64	White		26
5.7	74	White		28
6.1	62	Black		32

**Table 2 cancers-16-02735-t002:** Survey demographics.

Race	N (%)
Asian	26 (10.6)
Black	11 (4.5)
Native American	4 (1.6)
White	186 (75.9)
Unknown	18 (7.3)
**Ethnicity**	
Hispanic	52 (21.2)
Non-Hispanic	179 (73.1)
Unknown	14 (5.7)
**Family History of Diabetes**	
Yes	126 (51.4)
No	119 (48.6)
**US Born**	
Yes	48 (19.6)
No	197 (80.4)
**Health Status**	
Poor	2 (0.8)
Fair	19 (7.8)
Average	70 (28.6)
Good	119 (48.6)
Excellent	35 (14.3)

**Table 3 cancers-16-02735-t003:** Analysis of cellular senescence in T-cells from 22 women.

HbA1c	Age	Race	Ethnicity
**Metabolically Healthy**			
4.7	37	White	
4.9	53	White	
5.0	34	White	
5.0	53	White	Latina
5.2	52	White	
5.2	48	White	
5.3	51	White	Latina
5.4	47	Asian	
5.4	55	White	
5.5	48	White	
5.5	56	White	
5.5	43	Native American	
5.6	55	White	Latina
**Insulin Resistant**			
5.7	52	White	Latina
5.7	62	White	
5.7	61	Asian	
5.9	52	Black	
5.9	51	Black	
5.8	53	White	
6.3	45	Asian	
6.0	53	White	
6.3	51	White	Latina

**Table 4 cancers-16-02735-t004:** Methylation assessment of aging. There were 53 women evaluated. A total of 25 women were metabolically healthy, and 28 women were insulin-resistant; 29 women were aged ≤55, and 24 women were aged >55.

HbA1c	Age	Race	Ethnicity	Deprivation Index	BMI	Methylation Score
**Metabolically Healthy**						
4.7	37	White		2	22.9	33.4
4.9	53	White		3	29.7	48.4
5	53	White	Latina	4	30.5	43.1
5.2	48	White		2	25.9	38.9
5.3	55	Asian	Filipina	4	23.8	47.2
5.3	54	White	Latina	9	31.6	43.4
5.3	48	White		2	22.1	40.0
5.3	51	White	Latina	9	37.1	48.3
5.3	40	Asian		1	20.1	30.1
5.4	51	White		1	20.4	44.9
5.4	56	White		6	31	36.0
5.4	47	Asian		6	33.7	43.2
5.4	55	White		3	22	46.4
5.5	66	White		3	29.4	53.2
5.5	54	White		3	26.2	45.3
5.5	53	White		5	21.1	45.9
5.5	48	White		4	32.1	38.3
5.5	54	White		1	34.3	50.8
5.5	56	White		4	28.8	46.6
5.5	64	White		1	19	54.5
5.5	31	White	Latina	4	33.6	22.5
5.6	57	White		3	23.5	45.1
5.6	76	White		1	40.5	63.7
5.6	62	White		1	26.6	48.6
5.62	60	Black		3	20.3	44.8
**Insulin Resistant**						
5.7	51	Asian	Filipina	3	23.7	38.5
5.7	62	White		10	37.1	50.4
5.7	61	Asian		4	21	50.0
5.7	70	White		4	27.8	63.5
5.7	59	Black		8	24	43.7
5.8	56	White	Latina	4	32.9	50.9
5.8	74	White		5	31.3	56.7
5.8	47	Asian		4	21.8	37.5
5.8	49	White	Latina	9	32.8	32.7
5.8	46	White		6	23.6	38.5
5.8	50	White		5	24	59.1
5.8	69	White		9	33.8	51.5
5.8	53	White		3	35.9	43.1
5.8	83	White		7	21.6	58.3
5.8	61	White		10	33.3	52.7
5.9	63	White		2	29.2	53.9
5.9	59	White	Latina	7	29.2	45.4
5.9	49	White		8	23	42.5
5.9	52	Black		9	31.6	34.5
5.9	51	Black		2	38.2	36.6
5.9	56	White		3	30.8	48.5
6	74	White		4	36.6	60.6
6	54	White		5	35.3	38.3
6	72	Asian		2	19.6	59.7
6.1	57	Black		3	30.7	40.4
6.2	48	White	Latina	5	32.4	38.1
6.3	51	White	Latina	4	27.1	41.9
6.3	73	Asian		3	25.1	67.2

## Data Availability

De-identified ChIP seq data will be deposited at dbGap.

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
