# Peer review of "Insulin Resistance in Women Correlates with Chromatin Histone Lysine Acetylation, Inflammatory Signaling, and Accelerated Aging"

_cancers, 2024, doi:10.3390/cancers16152735_

Round 1
Reviewer 1 Report
Comments and Suggestions for Authors
General comment: The work is very interesting and well structured, also I appreciate the focus on female profiling, as we know gender medicine is a relatively new science and it tries to fill the biases of years of experimentation and research only on male cohorts. I congratulate the authors and advise to bring the data brought closer to reality, identifying the concrete boundaries of applicability of these studies.
Author Response
Response to Reviewer 1
Comment 1: I …advise to bring the data brought closer to reality, identifying the concrete boundaries of applicability of these studies.
Response 1: We greatly appreciate Reviewer 1’s comment. We identify the concrete boundaries and applicability of these studies in the revised manuscript in our new Conclusion.
Reviewer 2 Report
Comments and Suggestions for Authors
The article of Vidal Ch.M. et al represents an interesting example of the study aimed at elucidating the association between insulin resistance (IR), inflammation, aging and epigenetic changes (acetylation of histone H3K9ac). The material presented is physiologically important. The conclusions are quite convincingly confirmed experimentally.
At the same time the rank correlation between HgbA1c and individual cytokine’s values (Figure 5) represented as a 8 x 8 matrix is unsatisfactory and needs to be improved. It should be noted that only those results were obtained on a large sample of patients (245 women) and, at least for this reason, are especially valuable. Matrices of this type are usually denote the internal pairwise correlations between 8 different parameters, that is, in this case, between cytokines, but not between cytokines and HgbA1c – as it was indicated in the legend. And only as a result of pairwise comparison of cytokines can symmetry of the matrix appear – relative to its main diagonal, and all cells on the main diagonal will have correlation coefficient values ​​of “1”, which in fact can only occur in the case of pairwise comparisons of individual cytokines with each other. In addition, the diagonal symmetry assumes that all correlation values presented in the cells are duplicated, which also complicates the material presented – and so requires some additional comments. Also it is not clear what the authors meant by “High value” in the context of the names of cytokines marking the rows and columns of individual cells outside the matrix. This inscription is almost unreadable so may cause the wrong interpretation. In conclusion, I would like to note the unacceptably poor quality of the inscriptions in the figures, this especially applies to Fig. 5 and Fig. 2, where the inscriptions are practically unreadable.
Also there are several typos in the text: omissions or substitutions of the letters, e.g. “depravation index” (in Table 4 – instead of ‘deprivation’ – I would recommend to look up the meaning of ‘depravation’ in the dictionary), ‘ontrol’ instead of control group etc.
All negligence in the design regarding the necessary comments to Fig. 5, the clarity of the inscriptions in Fig. 5 and Fig. 2, as well as the typos in text must be corrected – and only after that a manuscript of such interesting content can be printed.
Comments on the Quality of English LanguageThere are several typos in the text: omissions or substitutions of the letters, e.g. “depravation index” (in Table 4 – instead of ‘deprivation’ – I would recommend to look up the meaning of ‘depravation’ in the dictionary), ‘ontrol’ instead of control group etc.
Author Response
Response to Reviewer 2:
Comment 1: Minor editing of English language is required.
Response 1: We apologize and have re-edited the manuscript.
Comment 2: At the same time the rank correlation between HgbA1c and individual cytokine’s values (Figure 5) represented as a 8 x 8 matrix is unsatisfactory and needs to be improved. It should be noted that only those results were obtained on a large sample of patients (245 women) and, at least for this reason, are especially valuable. Matrices of this type are usually denote the internal pairwise correlations between 8 different parameters, that is, in this case, between cytokines, but not between cytokines and HgbA1c – as it was indicated in the legend. And only as a result of pairwise comparison of cytokines can symmetry of the matrix appear – relative to its main diagonal, and all cells on the main diagonal will have correlation coefficient values of “1”, which in fact can only occur in the case of pairwise comparisons of individual cytokines with each other. In addition, the diagonal symmetry assumes that all correlation values presented in the cells are duplicated, which also complicates the material presented – and so requires some additional comments.
Response 2: We have revised Figure 5 to incorporate scatter plots of all combinations of variables on the lower triangle, distributions of each measure on the diagonal and Spearman correlations on the upper triangle. This provides more informative details of these measures over the population studied.
Comment 3: …it is not clear what the authors meant by “High value” in the context of the names of cytokines marking the rows and columns of individual cells outside the matrix.
This inscription is almost unreadable so may cause the wrong interpretation.
Response 3: This Figure is re-made integrating issues cited in Comment 2.
Comment 4: I would like to note the unacceptably poor quality of the inscriptions in the figures, this especially applies to Fig. 5 and Fig. 2, where the inscriptions are practically unreadable.
Response 4: We apologize. Fig. 2 now occupies a full page. Fig. 5 has been re-made in concordance with concerns raised in Comment 2.
Comment 5: …There are several typos in the text: omissions or substitutions of the letters, e.g. “depravation index” (in Table 4 – instead of ‘deprivation’ – I would recommend to look up the meaning of ‘depravation’ in the dictionary), ‘ontrol’ instead of control group etc.
Response 5: The manuscript has been re-edited. Please see Response 1.
Comment 6: All negligence in the design regarding the necessary comments to Fig. 5, the clarity of the inscriptions in Fig. 5 and Fig. 2, as well as the typos in text must be corrected – and only after that a manuscript of such interesting content can be printed.
Response 6: We thank Reviewer 2 for their comments and have revised the manuscript accordingly in Responses 1-5.